# XMixup: Efficient Transfer Learning with Auxiliary Samples by Cross-domain Mixup

## Abstract

Transferring knowledge from large source datasets is an effective way to fine-tune the deep neural networks of the target task with a small sample size. A great number of algorithms have been proposed to facilitate deep transfer learning, and these techniques could be generally categorized into two groups – *Regularized Learning* of the target task using models that have been pre-trained from source datasets, and *Multitask Learning* with both source and target datasets to train a shared backbone neural network. In this work, we aim to improve the multitask paradigm for deep transfer learning via *Cross-domain Mixup* (XMixup). While the existing multitask learning algorithms need to run backpropagation over both the source and target datasets and usually consume a higher gradient complexity, XMixup transfers the knowledge from source to target tasks more efficiently: for every class of the target task, XMixup selects the auxiliary samples from the source dataset and augments training samples via the simple mixup strategy. We evaluate XMixup over six real world transfer learning datasets. Experiment results show that XMixup improves the accuracy by 1.9% on average. Compared with other state-of-the-art transfer learning approaches, XMixup costs much less training time while still obtains higher accuracy.

## 1 Introduction

Performance of deep learning algorithms in real-world applications is often limited by the size of training datasets. Training a deep neural network (DNN) model with a small number of training samples usually leads to the *over-fitting* issue with poor generalization performance. A common yet effective solution is to train DNN models under transfer learning Pan et al. (2010) settings using large source datasets. The knowledge transfer from the source domain helps DNNs learn better features and acquire higher generalization performance for the pattern recognition in the target domain Donahue et al. (2014); Yim et al. (2017).

**Backgrounds.** For example, the paradigm Donahue et al. (2014) proposes to first train a DNN model using the large (and possibly irrelevant) source dataset (e.g. ImageNet), then uses the weights of the pre-trained model as the starting point of optimization and fine-tunes the model using the target dataset. In this way, blessed by the power of large source datasets, the fine-tuned model is usually capable of handling the target task with better generalization performance. Furthermore, authors in Yim et al. (2017); Li et al. (2018; 2019) propose transfer learning algorithms that regularize the training procedure using the pre-trained models, so as to constrain the divergence of the weights and feature maps between the pre-trained and fine-tuned DNN models. Later, the work Chen et al. (2019); Wan et al. (2019) introduces new algorithms that prevent the regularization from the hurts to transfer learning, where Chen et al. (2019) proposes to truncate the tail spectrum of the batch of gradients while Wan et al. (2019) proposes to truncate the ill-posed direction of the aggregated gradients.

In addition to the aforementioned strategies, a great number of methods have been proposed to transfer knowledge from the multi-task learning perspectives, such as Ge & Yu (2017b); Cui et al. (2018). More specifically, Seq-Train Cui et al. (2018) proposes a two phase approach, where the algorithm first picks up auxiliary samples from the source datasets with respect to the target task, then pre-train a model with the auxiliary samples and fine-tune the model using the target dataset. Moreover, Co-Train Ge & Yu (2017b) adopts a multi-task co-training approach to simultaneously train a shared backbone network using both source and target datasets and their corresponding separate

Fully-Connected (FC) layers. While all above algorithms enable knowledge transfer from source datasets to target tasks, they unfortunately perform poorly, sometimes, due to the critical technical issues as follows.

- **Catastrophic Forgetting and Negative Transfer.** Most transfer learning algorithms Donahue et al. (2014); Yim et al. (2017); Li et al. (2018; 2019) consist of two steps – pre-training and fine-tuning. Given the features that have been learned in the pre-trained models, either forgetting some good features during the fine-tuning process (*catastrophic forgetting*) Chen et al. (2019) or preserving the inappropriate features/filters to reject the knowledge from the target domain (*negative transfer*) Li et al. (2019); Wan et al. (2019) would hurt the performance of transfer learning. In this way, there might need a way to make compromises between the features learned from both source/target domains during the fine-tuning process, where multi-task learning with Seq-Train Cui et al. (2018) and Co-Train Ge & Yu (2017b) might suggest feasible solutions to well-balance the knowledge learned from the source/target domains, through fine-tuning the model with a selected set of auxiliary samples (rather than the whole source dataset) Cui et al. (2018) or alternatively learning the features from both domains during fine-tuning Ge & Yu (2017b).

- **Gradient Complexity for Seq-Train and Co-Train.** The deep transfer learning algorithms based on multi-task learning are ineffective. Though the pre-trained models based on some key datasets, such as ImageNet, are ubiquitously available for free, multi-tasking algorithms usually need additional steps for knowledge transfer. Prior to the fine-tuning procedure based on the target dataset, Seq-Train requires an additional step to select auxiliary samples and "mid-tunes" the pre-trained model using the selected auxiliary samples Cui et al. (2018). Furthermore, Co-Train Ge & Yu (2017b) requests additional cost for backpropagation in-situ as the two dataset combined. In this way, there might need a deep transfer learning algorithm that does not require explicit "mid-tuning" procedure or additional backpropagation to learn from the source dataset.

**Our Work.** With both technical issues in mind, we aim to study efficient and effective deep transfer learning algorithms with low computational complexity from the *multi-task* learning perspectives. We propose XMixup, namely *Cross-domain Mixup*, which is a novel deep transfer learning algorithm enabling knowledge transfer from source to target domains through the low-cost Mixup Zhang et al. (2018b). More specifically, given the source and target datasets for image classification tasks, XMixup runs deep transfer learning in two steps – (1) ***Auxiliary sample selection:*** XMixup pairs every class from the target dataset to a dedicated class in the source dataset, where the samples in the source class are considered as the auxiliary samples for the target class; then (2) ***Mixup with auxiliary samples and Fine-tuning:*** XMixup combines the samples from the paired classes of the two domains randomly using mixup strategy Zhang et al. (2018a), and performs fine-tuning process over the mixup data. To the best of our knowledge, this work has made three sets of contributions as follows.

1. We study the problems of cross-domain deep transfer learning for DNN classifiers from the multitask learning perspective, where the knowledge transfer from the source to the target tasks is considered as a co-training procedure of the shared DNN layers using the target dataset and auxiliary samples Ge & Yu (2017b); Cui et al. (2018). We review the existing solutions Donahue et al. (2014); Yim et al. (2017); Li et al. (2018; 2019), summarize the technical limitations of these algorithms, and particularly take care of the issues in catastrophic forgetting Chen et al. (2019), negative transfer Wan et al. (2019), and computational complexity.

2. In terms of methodologies, we extend the use of Mixup Zhang et al. (2018b) to the applications of cross-domain knowledge transfer, where both source and target datasets own different sets of classes and the aim of transfer learning is to adapt classes in the target domain. While vanilla mixup augments the training data with rich features and regularizes the stochastic training beyond empirical risk minimization (ERM), the proposed algorithm XMixup in this paper uses mixup to fuse the samples from source and target domains. In this way, the catastrophic forgetting issue could be solved in part, as the model keeps learning from both domains, but with lower cost compared to Chen et al. (2019). To control the effects of knowledge transfer, XMixup also offers a tuning parameter to make trade-off between the two domains in the mixup of samples Zhang et al. (2018b).

3. We carry out extensive experiments using a wide range of source and target datasets, and compare the results of XMixup with a number of baseline algorithms, including fine-tuning with weight decay ($L^2$) Donahue et al. (2014), fine-tuning with $L^2$-regularization on the starting point ($L^2$-$SP$) Li et al. (2018), Batch Singular Shrinkage (BSS) Chen et al. (2019), Seq-Train Cui et al. (2018), and Co-Train Ge & Yu (2017b). The experiment results showed that XMixup can outperform all these algorithms with significant improvement in both efficiency and effectiveness.

**Organizations of the Paper** The rest of this paper is organized as follows. In Section 2, we review the relations between our work to the existing algorithms, where the most relevant studies are discussed. We later present the algorithm design in Section 3, and the experiments with overall comparison results in Section 4, respectively. We discuss the details about the algorithm with case studies and ablation studies in Section 5, then conclude the paper in Section 6.

## 2 RELATED WORK

The most relevant studies to our algorithm are Donahue et al. (2014); Chen et al. (2019); Cui et al. (2018); Ge & Yu (2017b); Zhang et al. (2018b); Xu et al. (2020). All these algorithms, as well as the proposed XMixup algorithm, start the transfer learning from a pre-trained model, which has been well-trained using the source dataset. However, XMixup makes unique technical contributions in comparisons to these works.

Compared to Donahue et al. (2014), which fine-tunes the pre-trained model using the target set only and might cause the so-called catastrophic forgetting effects, XMixup proposes to fine-tune the pre-trained model using the mixup data from both domains. Compared Chen et al. (2019), which uses the computationally expensive singular value decomposition (SVD) on the batch gradients to avoid catastrophic forgetting and negative transfer effects, XMixup employs the low-cost mixup strategies to achieve similar goals. Compared to Cui et al. (2018), the proposed algorithm XMixup also adopts a similar procedure (pairing the classes in source/target domains) to pick up auxiliary samples from the source domain for knowledge transfer. XMixup however further mixes up the target training set with auxiliary samples and fine-tunes the pre-trained model with the data in an end-to-end manner, rather than using a two-step approach for fine-tuning Cui et al. (2018). Compared to Ge & Yu (2017b), which combines source/target tasks together to fine-tune the shared DNN backbones, the proposed algorithm here mixes up data from the two domains and boosts the performance through a simple fine-tuning process over the mixup data with low computational cost.

Finally, we extend the usage of vanilla mixup strategy Zhang et al. (2018b) for the applications of transfer learning, where in terms of methodologies we propose to pair classes of the two domains and perform mixup over the selected auxiliary samples for improved performance. Actually, mixup strategies have been used in Xu et al. (2020) for unsupervised domain adaption. Since the target task is assumed to share the same set of classes as the source domain in Xu et al. (2020), selecting auxiliary samples or pairing the source classes to fit the classes in the target domain is not required.

## 3 XMIXUP: CROSS-DOMAIN MIXUP FOR DEEP TRANSFER LEARNING

Given the source and target datasets and a pre-trained model (that has been well-trained using the source dataset), XMixup performs deep transfer learning using two steps as follows.

**Auxiliary Sample Selection**  Given a source dataset $S$ with $m$ classes and a target training dataset $T$ with $n$ classes, XMixup assumes the source domain is usually with more classes than the target one (i.e., $m > n$), and it intends to pair every class in the target training dataset with a unique and dedicated class in the source dataset (one-to-one pairing from the target to source classes). More specifically, given a pre-trained model, XMixup first passes every sample from the two datasets through the pre-trained model and obtains features extracted from the last layer of the feature extractor. Then, XMixup groups the features of the samples according to the ground truth classes in their datasets, and estimates the centroid of the features for every class in both datasets. Such that, for every class $c$ in the source or target dataset, XMixup represents the class as the centroid of the

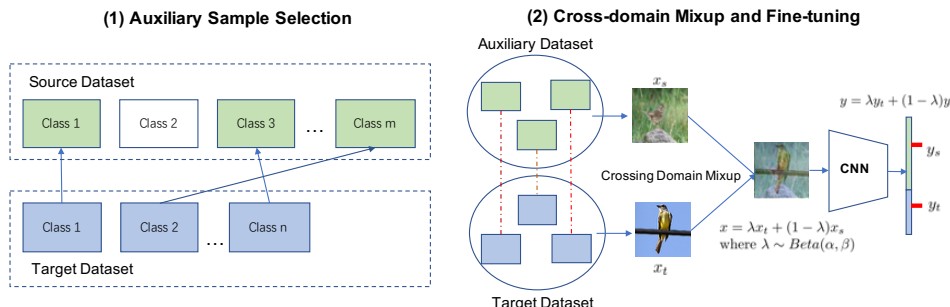

Figure 1: Overall Design of XMixup Algorithm

features using the pre-trained model $\Theta_{\text{pretrain}}$ for every sample $x_i$ in the class $c$, i.e.,

$$\text{centroid}(c) = \frac{1}{|c|} \sum_{\forall x_i \in c} \Phi(x_i, \Theta_{\text{pretrain}}), \text{ for } c \in S \text{ or } c \in T. \tag{1}$$

Given two classes $c_s$ and $c_t$ in the source and target domains respectively, we consider the similarity between the two classes as the potentials for knowledge transfer, while XMixup measures the similarity between the two classes using the *cosine measures* between the centroids of the two classes, such that $\text{dist}(c_s, c_t) = \text{cosine} < \text{centroid}(c_s), \text{centroid}(c_t) >$. In this way, the *auxiliary sample selection* could be reduced to search the optimal transport between the sets of classes of $S$ and $T$ respectively, via the pre-defined distance measure. Hereby XMixup intends to find a one-to-one mapping $\mathcal{P}^* : T \to S$, such that

$$\mathcal{P}^* \leftarrow \underset{\forall \mathcal{P} \subset (S \times T) \cap \text{O2O}}{\text{argmin}} \sum_{\forall c_t \in T} \text{dist}(c_t, P(c_t)), \tag{2}$$

where $S \times T$ refers to the Cartesian product of the target and source class sets, O2O refers to the constraint of the one-to-one mapping, $\mathcal{P}(c_t)$ maps the target class to a unique class from the source domain. Note that $\mathcal{P}^*$ refers to the optimal mapping that potentially exists to minimize the overall distances, while XMixup solves the optimization problem using a simple Greedy search Cui et al. (2018) to pursue a robust solution denoted as $\mathcal{P}^{\text{greedy}}$ in low complexity. Compared to XMixup, the Seq-Train algorithm Cui et al. (2018) uses Greedy algorithm to pair the source/target classes via the measure Earth Mover's Distance (EMD), which might be inappropriate in our settings of transfer learning.

**Cross-domain Mixup with Auxiliary Samples and Fine-tuning** Given the one-to-one pairing $\mathcal{P}^{\text{greedy}}$ from target to source classes, XMixup carries out the fine-tuning process over the two datasets. In every iteration of fine-tuning, XMixup first picks up a mini-batch of training samples $\mathcal{B}$ drawn from the target dataset $T$; then for every sample $x_t$ in the batch $\mathcal{B}$, the algorithm retrieves the class of $x_t$ as $x_t$.class and randomly draws one sample $x_s$ from the paired class of $x_t$.class, such that

$$x_s \overset{i.i.d}{\sim} \mathcal{P}^{\text{greedy}}(x_t.\text{class}), \ \forall x_i \in \mathcal{B}. \tag{3}$$

We consider $x_s$ as an auxiliary sample of $x_t$ in the current iteration of fine-tuning. XMixup then mixes up the two samples as well as their labels through linear combination with a trade-off parameter $\lambda$ drawn from the Beta distribution $Beta(\alpha, \beta)$, such that

$$x = \lambda x_t + (1 - \lambda)x_s, \ y = \lambda y_t + (1 - \lambda)y_s, \text{ and } \lambda \overset{i.i.d}{\sim} Beta(\alpha, \beta). \tag{4}$$

In this way, XMixup augments the original training sample $(x_t, y_t)$ from the target domain using the auxiliary sample $(x_s, y_s)$ from the paired source class, for knowledge transfer purposes. XMixup fine-tunes the pre-trained model $\Theta_{\text{pretrain}}$ using the mixup samples accordingly.

## 4 EXPERIMENTS AND OVERALL COMPARISONS

### 4.1 DATASETS

**Stanford Dogs.** The Stanford Dogs Khosla et al. (2011) dataset contains images of 120 breeds of dogs worldwide, each of which containing 100 examples for training and 72 for testing.

Table 1: Comparison of top-1 accuracy (%) with different transfer learning algorithms. $L^2$ refers to naive fine-tuning with weight decay. $L^2 - SP$ refers to regularizing around the pre-trained weight Li et al. (2018). $BSS$ refers to Batch Spectral Shrinkage Chen et al. (2019). Seq-Train refers to pre-training with the auxiliary dataset and then fine-tuning the target dataset Cui et al. (2018). Co-train refers to selective joint training with the auxiliary dataset Ge & Yu (2017a).

| Dataset | Regularized Learning | | | Multitask Learning | | |
|---|---|---|---|---|---|---|
| | $L^2$ | $L^2 - SP$ | $BSS$ | Seq-Train | Co-Train | XMixup |
| CUB-200-211 | 80.37±0.12 | 80.04±0.19 | 80.76±0.35 | 78.89±0.30 | 80.83±0.19 | 81.64±0.23 |
| Stanford Cars | 89.23±0.19 | 88.63±0.18 | 89.98±0.16 | 87.29±0.15 | 88.16±0.09 | 90.58±0.17 |
| Flower-102 | 91.64±0.32 | 91.98±0.24 | 91.59±0.42 | 90.27±0.31 | 90.66±0.33 | 93.45±0.14 |
| Food-101 | 83.84±0.10 | 84.06±0.04 | 83.90±0.06 | 83.49±0.09 | 84.04±0.14 | 84.33±0.05 |
| Stanford Dogs | 86.31±0.07 | 88.72±0.12 | 86.41±0.04 | 87.2±0.17 | 90.13±0.07 | 91.24±0.08 |
| FGVC-Aircraft | 83.12±0.60 | 82.51±0.38 | 83.77±0.63 | 80.74±0.49 | 82.78±0.42 | 84.66±0.57 |
| Average | 85.75 | 85.99 | 86.07 | 84.65 | 86.10 | 87.65 |

**CUB-200-2011.** Caltech-UCSD Birds-200-2011 Wah et al. (2011) consists of 11,788 images of 200 bird species. Each species is associated with a Wikipedia article.

**Food-101.** Food-101 Bossard et al. (2014) is a large scale data set consisting of more than 100k food images divided into 101 different kinds.

**Flower-102.** Flower-102 Nilsback & Zisserman (2008) consists of 102 flower categories. 1,020 images are used for training and 6149 images for testing. Only 10 samples are provided for each category during training.

**Stanford Cars.** The Stanford Cars Krause et al. (2013) dataset contains 16,185 images of 196 classes of cars. The data is split into 8,144 training and 8,041 testing images.

**FGVC-Aircraft.** FGVC-Aircraft Maji et al. (2013) is a fine-grained visual classification dataset composing more than 10,000 images of aircraft across 102 different aircraft models.

## 4.2 TRAINING DETAILS

We evaluate the recent state-of-the-art transfer learning methods in addition to XMixup. They are divided into two categories, which are regularized learning and multitask learning. For the former, we evaluate fine-tuning with $L^2$ regularization, $L^2 - SP$ regularization Li et al. (2018) and $BSS$ regularization Chen et al. (2019). Note that the multitask learning is defined in a broad sense here, including traditional co-training Ge & Yu (2017a), sequential training Cui et al. (2018) and co-training with XMixup. Compared with regularized learning, the essential difference is to re-train labeled auxiliary examples from the source dataset instead of regularizing the source model. We use the strategy described in Section 3 to select auxiliary examples for all multitask learning experiments. According to the empirical study in Ge & Yu (2017a), a threshold value of the auxiliary dataset size is required to guarantee the effect. In our implementation, we repeat the one-to-one pairing procedure until the size of the selected subset reaches a specific threshold. The value is 100,000 for Stanford Dogs and 200,000 for other datasets.

All experiments are performed using modern deep neural architecture ResNet-50 He et al. (2016), pre-trained with ImageNet. Input images are resized with the shorter edge being 256 and center cropped to square. We perform standard data augmentation composed of random flip and random cropped to $224 \times 224$. For optimization, we use SGD with the momentum of 0.9 and batch size of 48. We train 9,000 iterations for all datasets. The initial learning rate is set to 0.001 for Stanford Dogs and 0.01 for remaining datasets. It is divided by 10 after 6,000 iterations. Each experiment is repeated five times. The average top-1 classification accuracy and standard division are reported.

## 4.3 RESULTS

**Accuracy.** The top-1 classification accuracies are reported in Table 1. We observe that regularized learning methods obtain similar results, while $L^2 - SP$ performs obviously better on Stanford Dogs. $BSS$ achieves similar improvements on average but is more stable, consistently outperforming $L^2$

Table 2: Evaluating the accuracy on different subsets of the source dataset. They are samples in the auxiliary dataset (Auxiliary), samples not in the auxiliary dataset (ABA: All But Auxiliary Samples), and the whole source dataset (All).

| Subset | CUB-200-2011 | | FGVC-Aircraft | | Stanford Dogs | |
|--------|-------|--------|-------|--------|-------|--------|
| | $L^2$ | XMixup | $L^2$ | XMixup | $L^2$ | XMixup |
| Auxiliary | 62.11 | 62.91 | 49.76 | 52.78 | 75.92 | 79.37 |
| ABA | 38.17 | 36.82 | 20.36 | 18.93 | 64.67 | 64.84 |
| All | 38.73 | 36.83 | 26.40 | 26.89 | 64.39 | 64.70 |

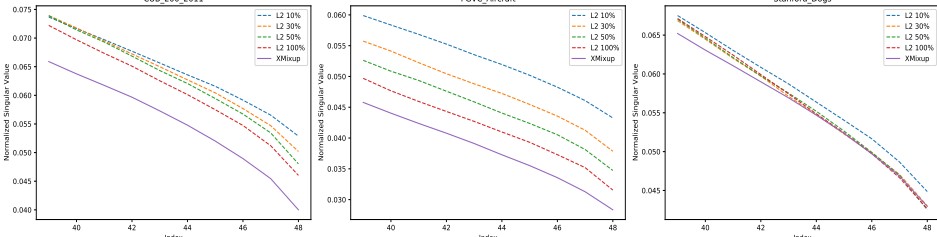

Figure 2: Singular values of feature matrices extracted by different transfer learning models. Singular values are divided by the corresponding largest one for scale normalization. Top 10 smallest values are presented. $L^2$ models are trained with random sampling rates 10%, 30%, 50% and 100% respectively.

by a small margin on most datasets. As for multitask learning, pre-training with auxiliary examples before fine-tuning often hurts the performance surprisingly. This may be caused by the phenomenon of catastrophic forgetting Li & Hoiem (2017), that pre-training with a subset of the source dataset loses general knowledge stored in the source model, but this general knowledge is probably useful for the target task. Co-training with auxiliary examples performs much better because auxiliary examples from the source task are used to preserve the knowledge. However, Co-training still hurts the performance on some datasets which have larger distance of data distribution to the source dataset such as Stanford Cars and FGVC-Aircraft. XMixup obtains obvious higher average accuracy than all baseline methods. It is also very robust and stably outperforms naive fine-tuning with $L^2$ regularization.

**Complexity.** As stated in Li et al. (2018); Chen et al. (2019), regularized learning methods are usually efficient because the computational complexity involved by the regularization is approximately proportional to the number of parameters or features. While previous multitask learning methods Ge & Yu (2017a); Cui et al. (2018) are time consuming to deal with auxiliary examples, costing additional time whose scale is at least the same as training only target examples. XMixup achieves the efficiency almost the same as naive fine-tuning, because extra computations for data mixing are negligible.

## 5 DISCUSSIONS AND ABLATION STUDIES

In this section, we provide more empirical studies to analyze the effectiveness and applicability of our algorithm. In subsection 5.1, we show that XMixup effectively alleviates catastrophic forgetting and negative transfer. In subsection 5.2 and 5.3, XMixup is proved to be not sensitive with the auxiliary dataset selection and hyperparameter setting, indicating that XMixup is easy to be applied in various real world tasks. Finally in subsection 5.4, we analyze two essential characteristics, *crossing domain* and *supervised mixing*, showing that they are both necessary.

### 5.1 ANALYSIS OF THE EFFECTIVENESS

Authors in Chen et al. (2019) figure out that deep transfer learning suffers from two kinds of problems, which are *catastrophic forgetting* and *negative transfer*. We do empirical analysis showing that XMixup explicitly or implicitly deals with both issues. Three different datasets are used for detailed analysis.

**Catastrophic Forgetting.** A widely adopted measurement about catastrophic forgetting is to evaluate the accuracy of a fine-tuned model on the previous task Li & Hoiem (2017); Kirkpatrick et al.

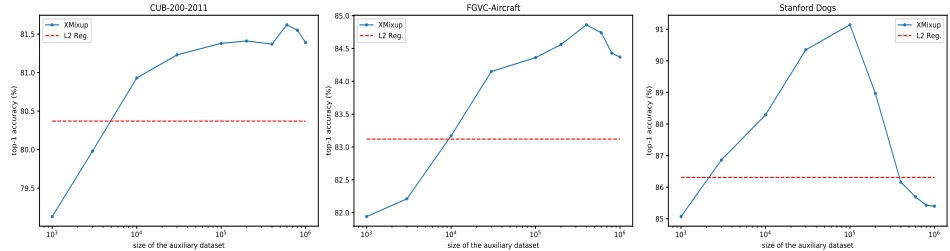

Figure 3: Influence of the auxiliary dataset size on the performance of XMixup.

Table 3: Top-1 accuracy of XMixup using random auxiliary examples.

| CUB-200-2011 | Stanford Cars | Flower-102 | Food-101 | Stanford Dogs | FGVC-Aircraft |
|---|---|---|---|---|---|
| 81.39(-0.25) | 90.66(+0.08) | 93.33(-0.12) | 84.12(-0.21) | 85.58(-5.66) | 84.37(-0.29) |

(2017). It is worth noting that although parameters are totally changed after fine-tuning, the general knowledge is still preserved for feature representation. We thus can use models fine-tuned on target tasks as feature extractors for images in source domains. A randomly initialized fully connected layer is trained to adapt to each specified source task. Results in Table 2 show that XMixup helps preserve the knowledge about auxiliary samples, although this may hurt the capacity of representing samples not in the auxiliary dataset.

**Negative Transfer.** Chen et al. (2019) finds that the distribution of tail singular values indicates the degree of negative transfer. Specifically, negative transfer can be reduced by suspending tail singular values. In Figure 2, we observe that XMixup shows similar trends with increasing number of training examples, which is the most meaningful approach to avoid negative transfer. Through sufficient utilization of auxiliary examples, XMixup further decreases smallest singular values.

## 5.2 SENSITIVITY OF THE AUXILIARY DATASET

Since the only external dependence is the auxiliary dataset, we analyze in detail how characteristics of the auxiliary dataset influences the effect of XMixup.

**Size.** We first discuss how XMixup performs as the size of the auxiliary dataset varies. We create auxiliary datasets with different scales based on the default dataset described in experiment settings. To increase the size, we continue adding the most similar category from the source dataset until the whole set is selected. To decrease the size, we perform random sampling from the default dataset. Note that these smaller auxiliary datasets are only different on the size but not the domain similarity, while larger auxiliary datasets are less similar with the target domain. As illustrated in Figure 3, CUB-200-2011 and FGVC-Aircraft show very good robustness to the size of the auxiliary dataset. Specifically, XMixup performs well enough when the number of auxiliary examples is more than 100,000. Surprisingly, we find that XMixup still significantly outperforms naive fine-tuning even simply using the whole source dataset without selection. The task of Stanford Dogs is a bit different that using too many dissimilar auxiliary examples hurts the performance, since it is closely related with a subset of ImageNet. However, all tasks can benefit from XMixup obviously with a wide range of the number of auxiliary examples starting from about 30,000.

**Domain Similarity.** In order to investigate how XMixup depends on the similarity of auxiliary examples, we keep the size of the auxiliary dataset the same, but only replace auxiliary examples by random sampling from the entire source dataset. The result is presented in Table 3. We observe that most datasets are not affected obviously by the similarity removing, indicating that XMixup is a robust approach to integrate the general knowledge during fine-tuning. In subsection 5.4, we further show through ablation study that, knowledge from the source domain plays an important role in XMixup. In other words, crossing domain mixing is more than a kind of perturbation on target examples.

Table 4: XMixup without the characteristic of crossing domain (Mixup) or supervised mixing (w/o Label) are evaluated respectively.

| CUB-200-2011 | | | FGVC-Aircraft | | | Stanford Dogs | | |
|---|---|---|---|---|---|---|---|---|
| XMixup | Mixup | w/o Label | XMixup | Mixup | w/o Label | XMixup | Mixup | w/o Label |
| 81.64 | 80.21 | 77.89 | 84.66 | 82.79 | 80.40 | 91.24 | 86.43 | 83.65 |

## 5.3 Sensitivity of Hyperparameters

There are two parameters $\alpha$ and $\beta$ in Beta distribution. We use fixed $\beta$ of 1 and only change $\alpha$ to control the mixing weight of the two domains. Larger $\alpha$ means larger sampling weights for examples from the target domain. We explore a broad range of values for $\alpha$ and investigate how well XMixup performs under different settings. We illustrate Beta distribution and the influence of varying $\alpha$ in Fig. 4 and Fig. 5 in Appendix.

**General trends.** We observe that, in all experiments conducted, XMixup demonstrates a relatively continuous and smooth change as $\alpha$ varies. Top-1 accuracy tends to be lower when $\alpha$ is small because the training assigns a too small weight to samples in the target dataset. It then rises to its maximum as $\alpha$ increases, followed by a gradual drop when $\alpha$ continues to increase as using a too large value for $\alpha$ degenerates to naive fine-tuning.

**Suggested range for $\alpha$ depending on datasets.** Our experimental results show that, for datasets that are very similar to the source datasets, such as Stanford Dogs which is a subset of ImageNet, a lower $\alpha$ ($2^{-3} \sim 2^{-1}$) usually performs better. Otherwise, $\alpha$ in a wide range between $2^0$ and $2^6$ is generally safe to improve the naive fine-tuning and even state-of-the-art baseline methods.

## 5.4 Ablation Study

To further determine whether both components, namely *crossing domain* and *supervised mixing*, are essential for XMixup, we evaluate the two variants. First we remove the characteristic of crossing domain, implementing traditional Mixup Zhang et al. (2018b) using only samples in the target dataset (in-domain). Second, we keep crossing domain sample mixing in the data generation step but remove labels of auxiliary samples .

As reported in Table 4, Mixup has lower test accuracy than XMixup by an average of 2.7% in the three datasets we experimented. This consistent difference implies that crossing domain is an essential factor that contributes to the effectiveness of XMixup. Furthermore, Mixup has similar, or even slightly worse, performance compared to $L^2$ models, whose results are shown in Table. 1. This phenomenon suggests that general data augmentation may not work well under transfer learning scenario where sample size is limited. Table 4 also shows that XMixup is more than data augmentation. We observe that model performance drops significantly (for an average of 5.2%) if labels of auxiliary examples are removed. This result indicates that knowledge of the source dataset is essential.

## 6 Conclusions

In this paper, we study the problem of knowledge transfer for DNN models from the perspectives of multi-task learning. We propose a novel deep transfer learning algorithm XMixup, namely Cross-domain Mixup, with superiority in both effectiveness and efficiency. Through a two-step approach with (1) auxiliary sample selection and (2) cross-domain mixup and fine-tuning, XMixup achieves significant performance improvements with 1.9% higher classification accuracy on average, when compared to the state-of-the-art algorithms Donahue et al. (2014); Li et al. (2018); Chen et al. (2019). XMixup is also robust to hyperparameter choices and ways of auxiliary sample selection. Finally, we conclude that knowledge transfer through multitask learning with a set of selected auxiliary samples is no doubt a promising direction with huge potentials, while this work suggests a solid yet easy-to-use baseline method.

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

# A    APPENDIX

## A.1    BETA DISTRIBUTION

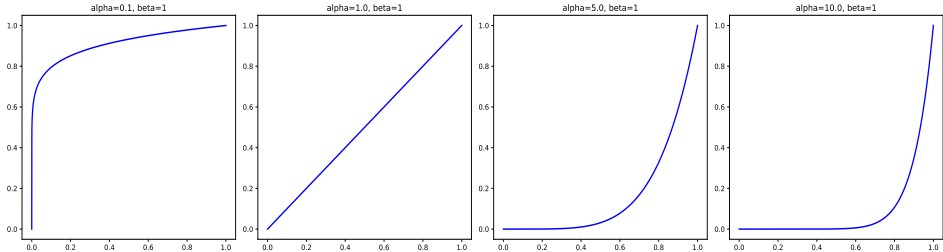

Figure 4: Examples of Beta Distribution.

## A.2    INFLUENCE OF THE MIXING WEIGHT

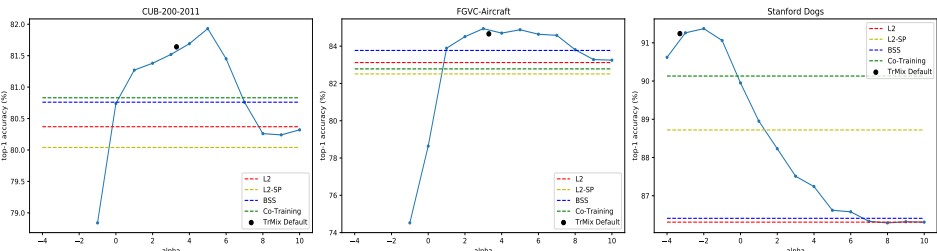

Figure 5: Influences of the choice of the hyperparameter $\alpha$ in log scale. Black nodes refer to the default value used in previous experiments. Doted lines refer to the accuracy of state-of-the-art baselines.

