# OpenReview forum: "XMixup: Efficient Transfer Learning with Auxiliary Samples by Cross-Domain Mixup"
_ICLR.cc/2021/Conference — Reject_

### Official Review · AnonReviewer4 · 2020-10-24
**Limited novelty and marginal improvements**

**Rating:** 4
**Confidence:** 5

**Review:**

Summary:

This paper proposes a simple variant for the mixup training mechanism for transfer learning problems: cross-domain mixup (XMixup). The key idea is to mix up the training samples from both domains where the samples are generated by nearest-center assignment in each class. Experiments on several datasets have shown its effectiveness in transfer learning compared to some SOTA methods.

Pros:

1. A simple but effective method for mixup in transfer learning.
2. The method is easy to implement with improvements in experiments.
3. The experiments are fair enough.

Cons:

1. The main disadvantage is its limited novelty. Computing the class center of each class and then initiating an assignment between each source and target class is a common strategy and also has been used by several existing works (see references below). In addition, the experiments on arbitrary sample selection (table 3) also show that even using random samples to form the mixup training samples, the performance is also good and has a very small gap to that in table 1. Therefore, I think this method can only be seen as a moderate extension to mixup for transfer learning problems.
2. The second disadvantage is that even if the experiments in table 1 show some positive results, the improvement is rather marginal, and all the datasets are rather small. For example, the largest dataset in this paper is Stanford Cars with 16,185 images where XMixup only achieves an accuracy improvement of 0.6%, which is not significant. Why not use other larger datasets such as Caltech-101, Caltech-256, Sun397, STL, and CIFAR-100? Since the method itself is simple, you should experiment on more larger datasets to show that it really works.
3. This method is problematic in real applications. It is not easy to use as it requires to store and access the source domain data (e.g., ImageNet), which is almost not possible. Therefore, even if XMixup is simple and doable in the idea level, the efficiency should be considered by not only the running time, but also the data storage and access cost. To this end, the regularized learning methods in table 1 are more applicable since they do not have to access the original source data. In addition, only marginal improvements are achieved by storing and accessing the large volume of source data. Therefore, the method in its current form is not helpful in real applications.
4. One of the claims in this paper is that XMixup can prevent catastrophic forgetting, i.e., "remember" both the source and target tasks. Why do we care about the forgetting phenomenon in transfer learning since its goal is to transfer the knowledge to the target domain to ensure that best performance can be achieved in the target domain? Do you have any evidence to show that preventing the catastrophic forgetting can somehow benefit the performance on the target?
5. Continued from point four, the efficiency of this method is not validated in experiments to compare with both regularized learning and multitask learning methods. So, it is hard to evaluate its time efficiency.
6. Finally, there lacks detailed analysis of why this simple mixup strategy would work apart from the accuracy results. For instance, what kind of representations did this method learn by mixing up that help it generalize better to target domain? Why it achieves comparable results with random selection?
7. Application scenario: All images in this paper are natural images that should remain similar in general. Will this method work for distant domain images such as ImageNet->medical images? It remains unclear when to use this method. A doable application scenario is needed.

To sum up, this is a good paper, but not appropriate for ICLR in its current form due to limited novelty, poor analysis, and marginal improvements.

References:

[1] Snell J, Swersky K, Zemel R. Prototypical networks for few-shot learning[C]//Advances in neural information processing systems. 2017: 4077-4087.

[2] Pan Y, Yao T, Li Y, et al. Transferrable prototypical networks for unsupervised domain adaptation[C]//Proceedings of the IEEE Conference on Computer Vision and Pattern Recognition. 2019: 2239-2247.

[3] Wang J, Chen Y, Yu H, et al. Easy transfer learning by exploiting intra-domain structures[C]//2019 IEEE International Conference on Multimedia and Expo (ICME). IEEE, 2019: 1210-1215.

[4] Hu D, Liang J, Hou Q, et al. PANDA: Prototypical Unsupervised Domain Adaptation[J]. arXiv preprint arXiv:2003.13274, 2020.

[5] Sun Q, Liu Y, Chua T S, et al. Meta-transfer learning for few-shot learning[C]//Proceedings of the IEEE conference on computer vision and pattern recognition. 2019: 403-412.

[6] Choi J, Hwang S J, Sigal L, et al. Knowledge Transfer with Interactive Learning of Semantic Relationships[C]//AAAI. 2016: 1505-1511.

---

> ### Author Response · Authors · 2020-11-25
> **Thank you for your comments**
>
> Thank you for your review and constructive comments.  We believe all your comments could greatly help us improve the manuscript.  Here we would like to clarify the seven concerns listed in your comments.
>
> 1.	This work is motivated by inventing a practical transfer learning strategy to fine-tune a pre-trained source model. Although some effective algorithms have been proposed, they are far from being satisfied especially for their inefficiency. Our work intends to tackle this problem by incorporating Mixup. As you pointed out, Mixup has already been successfully applied to many related machine learning tasks such as few-shot learning, unsupervised domain adaptation and semi-supervised learning. However, they are different types of transfer learning problems. Specifically, our work belongs to inductive transfer learning where labeled target examples are available (usually not quite sufficient nor very few). From the perspective of introducing a successful approach of transfer learning, incorporating fine-tuning with subset selection and Mixup makes sense and needs validation. Although these techniques are actually not novel, our purpose is to contribute to solve an important real world problem. Moreover, we observe that even randomly selected auxiliary samples can help the target task to generalize. We don’t agree that such a result hurts our contribution, and conversely, this demonstrates the potential of using a more simpler implementation (without auxiliary subset selection) of our method.
>
> 2.	The improvements may not be significant if measured by the absolute value of the accuracy. While these numbers are reasonable compared with other state-of-the-art works. Furthermore, we obtain stable improvements over all evaluated datasets. Large datasets are usually not employed as most inductive transfer learning works[1,2,3,4] do, since the benefit from transfer learning may be marginal (e.g. [5] give the empirical experience that ImageNet pre-trained model does not necessarily improves accuracy unless the target dataset contains fewer than 10k images). Furthermore, tasks with fewer than 10k training images are very common in practice due to the high cost on sample labeling.
>
> 3.	We admit that the requirement of extra storage for auxiliary samples is a disadvantage of our method, and actually all transfer learning methods involving auxiliary samples like Seq-train and Co-train. However, we aruge that the volume of required storage (e.g. ~100G bytes disk for ImageNet) is acceptable in many scenarios. Moreover, in applications of deep learning platforms, the cost is even more negligible as these general images can be shared among many users.
>
> 4.	The benefit of preventing the catastrophic forgetting is only theoretically discussed in previous transfer learning studies. While we would like to provide more empirical evidences to show its necessity.
>
> 5.	We have analyzed the theoretical time complexity, mainly compared with multitask learning methods. We believe that empirical results will not make remarkable difference as the analysis is almost hardware independent.
>
> 6.	In this current paper, we do lack detailed discussion about why our method is effective, especially when compared with other multitask learning methods. Also, experiments for transferring between distant domains such as ImageNet->medical images are useful to validate our methods. We are going to solve these problems in the next version.
>
> [1] Borrowing treasures from the wealthy: Deep transfer learning through selective joint fine-tuning. In Proceedings of the IEEE conference on computer vision and pattern recognition, pp. 1086–1095, 2017a.
>
> [2] Explicit inductive bias for transfer learning with convolutional networks. Thirty-fifth International Conference on Machine Learning, 2018.
>
> [3] DELTA: Deep learning transfer using feature map with attention for convolutional networks. In International Conference on Learning Representations, 2018b.
>
> [4] Catastrophic forgetting meets negative transfer: Batch spectral shrinkage for safe transfer learning. In Advances in Neural Information Processing Systems, pp. 1906–1916, 2019.
>
> [5] 	Rethinking ImageNet Pre-training. In Proceedings of the IEEE international conference on computer vision (pp. 4918-4927).

---

### Official Review · AnonReviewer3 · 2020-10-28
**Simple and moderately effective method for transfer learning, but not quite ready for publication**

**Rating:** 5
**Confidence:** 4

**Review:**

This paper proposes XMixup, a strategy for improving transfer learning in neural networks. Specifically, XMixup consists of mixup applied between target samples and source samples from the class pre-determined to be closest to target sample’s class. Experiments conducting transfer learning from pre-trained ImageNet to 6 smaller image classification datasets demonstrate XMixup to outperform the baseline approaches.

Pros:
1.	The proposed method is conceptually simple and easy to understand/implement. Since source and target information are combined into a single sample, only a single gradient needs to be backpropagated through the network. Additionally, the mixup operation is simple (draw from beta distribution, then a convex combination of the two images + labels), so overall XMixup is fairly cheap computationally, once class feature centroids have been computed (see Cons #2).
2.	The experiments demonstrate that XMixUp is fairly effective for the target image classification datasets shown, moderately outperforming the baseline methods on the 6 datasets, which have varying degree of overlap/similarity with ImageNet (source dataset).
3.	The authors made an effort to demonstrate why XMixup is effective, with many different kinds of experiments, including several I wouldn’t have thought of. See Cons #1, however.

Cons:
1.	The novelty of XMixup is moderately low; for the most part, it’s a fairly standard application of mixup, applied to transfer learning with a simple pairing of source and target classes. This could be fine if the experiments were particularly thorough or convincing. However, all experiments are in image classification, transferring from ImageNet as the source dataset. I would have liked to see experiments on more settings (e.g. NLP) or problem types (e.g. source: classification -> target: object detection). Additionally, several ablation studies that would shed more light on how XMixup works (see 2. below) are missing.
2.	While training is fairly efficient, XMixup does require computing the centroids for all source and target samples first, which can be somewhat expensive (for example, if ImageNet pre-training is the source, then features for all 1M ImageNet samples must be computed). It’s not clear how important this class-matching step is though. An ablation study without class matching target samples with the closest source class would be helpful. It’s also unclear how essential that the mapping between source and target classes be one-to-one; again, an ablation would have been helpful.
3.	One of the central arguments made is that XMixup prevents catastrophic forgetting, with experiments (Table 2) providing further evidence. Why does catastrophic forgetting matter for transfer learning though? “Forgotten” features are most likely to be those from the source pre-training task that have little relevance to the target task. For example, if fine-tuning ImageNet for aircraft classification, why should one care if the network can still distinguish dog breeds, or foods?
4.	The writing quality is not up to publication standards; please give this draft another careful round of edits. A non-exhaustive list of corrections:
o	Pg 1: Backgrounds -> background
o	“Later, the work Chen et al. (2019); Wan et al. (2019) introduces new algorithms that prevent the regularization from the hurts to transfer learning.”
o	“two phase” -> “two-phase”
o	“large scale” -> “large-scale”
o	Both “pre-train” and “pretrain” appear in the text
o	“each of which containing 100 examples” -> “each containing 100 examples”
o	“standard division” -> “standard deviation”

Questions:
-	Using centroids of the features for each class (Eq 1) to determine class similarity makes several assumptions about the shapes of the class distributions in the feature space. Have you inspected the feature space of the pre-trained network to verify if these assumptions are reasonable?
-	What is the dimensionality of the model output? Assuming m source classes and n target classes, is the output dimensionality m, or m+n? Figure 1 implied the latter, but I didn’t see it stated anywhere in the text.
-	Figure 3: Why would there be a drop in performance (especially for Stanford dogs) when the auxiliary dataset gets large? Why is it surprising that XMixup would outperform fine-tuning when using the full source dataset?
-	Pre-training is not only used just for classification. Object Detection, segmentation, VQA, and many other types of problems also rely on pre-trained networks. Can XMixup be used for other problem types? What about for NLP (e.g. BERT fine-tuning)?

Miscellaneous:
-	Unless using it like a noun, citations should be parenthetical (\citep).
-	x_t.class is awkward notation
-	Table 1: It would be useful to see plain naïve fine-tuning (no regularization, no XMixup, etc.) as well. Also, consider bolding the best values; makes the table easier to read.

Decision:
Despite mediocre novelty, XMixup seems to be an effective and simple technique. However, although the authors include several experiments aimed at understanding XMixup’s efficacy, several key ablations are missing, and there’s a lack of diversity in experimental settings (all ImageNet -> smaller image classification set). The writing is also not quite at publication standards and could use some work. As such, while I think there may be potential, I don’t think this work is ready at this time, and vote for rejection. I think this work can be significantly improved with increased experimental setting variety, better ablations, and a careful round of edits.


=======
Post-rebuttal
=======

I thank the authors for their comments. The current draft isn't a bad start, but in agreement with the other reviewers, it needs more work before it's ready for publication. Please carefully consider our recommendations for your next draft.

---

> ### Author Response · Authors · 2020-11-25
> **Thank you for your comments**
>
> Thank you for your encouraging review and constructive comments.  We believe all your comments could greatly help us improve the manuscript.  Here we would like to answer the four questions listed in the comment.
>
> 1. We have not inspected the feature space as we think it’s rather difficult to improve the subset selection strategy according to the shapes of class distributions. Another reason is that,  in many cases the benefit is rather obvious even we use random selected auxiliary examples for the cross-domain Mixup.
>
> 2. The dimensionality is m+n, as we do not assume the overlap between the source and target classes . We will add a clear statement in the next version.
>
> 3. Intuitively, an auxiliary dataset could benefit the target dataset (no matter in which manner they interact with each other, e.g. by XMixup or seq/co-training) if they are similar, otherwise the auxiliary dataset may hurt the target task which leads to the phenomenon of negative transfer. Therefore it is not surprising that the performance of the target task decreases as the auxiliary dataset become larger, since more irrelevant samples are included. Specifically, the source dataset ImageNet consists of various categories of objects such as animals, plants, tools and vehicles. It’s a bit counter-intuitive that a random selected subsets from such diverse kinds of objects can significantly help the task in a narrow area such as the bird or car.
>
> 4. The method is able to be applied in other problem types including object detection, segmentation, NLP and so on. We will validate our method in more types in future.
>
> We also would like to clarify a concern proposed in the 3rd item of Cons about “why does catastrophic forgetting matter for transfer learning”. The source model well trained over a large scale dataset contains useful knowledge which is helpful for many down-streaming tasks. For examples, good representations about the head or fur can be learned by various kinds of animals in the source dataset. Without proper regularization (through weights, features or samples), the general knowledge may be forgotten by over-fitting the target dataset. The problem is severer if the target examples are insufficient. Therefore, many transfer learning works make effort to tackle the problem of selecting a target-specific subset of samples or knowledge to preserve the relevant knowledge contained in the source task.
>
> We also thank you very much for the detailed review. Matters about the notation and citation will be handled in the next version.

---

### Official Review · AnonReviewer2 · 2020-10-28
**Method has limited technical novelty and it is not well motivated.**

**Rating:** 4
**Confidence:** 4

**Review:**

Summary:

This paper proposes a method to address the transfer learning in the multi-task learning setup. The proposed method has two main components: (1) finding a one-to-one mapping from source dataset classes to the target dataset classes (2) applying mixup between the samples from the source and target class pairs.

Reasons for score:
Overall, I vote for rejecting the paper.

1.The main idea of this paper is to compute the one-to-one pairing of classes in source and target domain ( i.e. for a given class in target dataset, find the closest class in the source dataset) I think this idea of having one-to-one mapping is flawed in principle. For example, let us consider that  the source dataset consists of 100 classes which represent different dog breeds and the target dataset has 10 classes representing different cars. In this scenario, one to one pairing between a dog class and the car class has no benefit for transfer learning. The authors should clearly state the underlying assumptions about the source dataset and the target datasets for the proposed method to work.

2. The technical novelty of the paper is very limited. The closest related work to this paper is Cui et. al. which selects K classes from source domain which are closest to the target dataset classes. In this work, the authors propose instead to use one-to-one pairing, without explicit explanation of why this is a good idea. Applying mixup between the source and target dataset samples is not novel by itself, without making explicit why it helps to facilitate better transfer learning.

3.The paper claims that “XMixup however further mixes up the target training set with
auxiliary samples and fine-tunes the pre-trained model with the data in an end-to-end manner, rather than using a two-step approach for fine-tuning Cui et al. (2018).” As far as I understand this is the biggest contribution of this paper ( to make the learning efficient), yet the authors do not make an attempt to elaborate  on this point of “two-step approach of Cui et. al vs their approach)

4.Overall, the tone of the paper is not scientific at many places ( examples below)

Pros:
The paper studies a very relevant problem of transfer learning.
It seems that the authors have tried to do robust experimental evaluation.

Other minor comments:

(A)The sentence formation is incorrect at many places:
“Later, the work Chen et al. (2019);Wan et al. (2019) introduces new algorithms that prevent the regularization from the hurts to transfer learning,”
“Given two classes c s and c t in the source and target domains respectively, we consider the similarity between the two classes as the potentials for knowledge transfer, while XMixup measures the similarity between the two classes using the cosine measures between the centroids of the two classes”
“Note that P ∗ refers to the optimal mapping that potentially exists to minimize the overall distances, while XMixup solves the optimization problem using a simple Greedy search”

(B)“While vanilla mixup augments the training data with rich features and regularizes
the stochastic training beyond empirical risk minimization (ERM),” saying “rich features” is not a technical description of the related work.

C. ”According to the empirical study in Ge & Yu (2017a), a threshold value of the auxiliary dataset size is required to guarantee the effect. In our implementation, we repeat the one-to-one pairing procedure until the size of the selected subset reaches a specific threshold. The value is 100,000 for Stanford Dogs and 200,000 for other datasets. ” This seems like an important hyperparameter, yet it is not obvious where it is in the method section.

---

> ### Author Response · Authors · 2020-11-25
> **Thank you for your comments**
>
> Thank you for your review and constructive comments.  We believe all your comments could greatly help us improve the manuscript.  Here we would like to clarify some major concerns in your comments.
>
> 1.	The underlying assumptions about the source and target datasets.
> We agree that we should have clearly stated the assumptions under which our method makes sense. Indeed, our method is suitable for the scenario that the source dataset is large and designed for general purpose. A typical case is the ImageNet dataset, which contains 1000 kinds of objects. The target task often focuses on one or a few specific domain(s) which is common in practice. Therefore, many transfer learning works make effort to tackle the problem of selecting a target-specific subset of samples or knowledge to help the target task generalize.
>
> 2.	The explanation about using one-to-one pairing.
> The greedy one-to-one paring strategy is intuitive and easy to implement. Cui et.al. describes that they impose the EMD to calculate the distance between two domains (either of which contains multiple categories) and then adopts a one-to-one greedy strategy to select the auxiliary categories. Actually we do not quite understand how the EMD distance helps the selection together with the one-to-one greedy manner.
>
> 3.	Elaborating on the point of the effectiveness of our end-to-end method.
> The reason why XMixup facilitates better transfer learning is not clearly demonstrated in our paper. We would try to resolve the problem in future versions.
>
> 4.	Thank you very much for all the corrections about our presentations.

---

### Official Review · AnonReviewer1 · 2020-11-03
**Recommend to Reject due to Limited Novelty and Missing Comparisons**

**Rating:** 4
**Confidence:** 5

**Review:**

This paper presents a multi-task paradigm for deep transfer learning via cross-domain mixup. Specifically, authors develop a strategy to transfer the knowledge from source to target tasks more efficiently by selecting the auxiliary samples from the source
dataset and augments training samples via the simple mixup strategy. The paper is well written and easy to follow. Experiments on six datasets show the efficacy of the proposed method over existing alternatives.

Overall, I vote for rejecting the paper due to limited novelty and lack of convincing experiments. The proposed approach simply combines different existing well-known techniques such as sample selection and mixup without any significant changes. Mixup has been widely used as an augmentation technique to improve the performance of a model. Similarly, as mentioned by the authors, source sample selection is also not new in transfer learning and has shown to be effective in Seq-Train. The only difference with Seq-Train is that the proposed approach uses a multi-task learning set up instead of a two phase approach for transfer learning. Low-cost mixup has also been proposed in Zhang et. al, 2018. Thus, I fail to understand the major contributions of the paper beside combination of the two simple strategies for transfer learning.

Authors mention about reviewing exiting solutions as a contribution in the paper. However, apart from a couple of paragraphs, there is no analysis on the catastrophic forgetting, negative transfer and efficiency. How does the proposed method is superior in terms of catastrophic forgetting and negative transfer compared to prior multi-task learning approaches? How prior methods are affected by catastrophic forgetting is not clear? More experiments and analysis is essential to consider this as a major contribution of the paper.

Mixup across different domain samples has been used. in domain adaptation. E.g., Adversarial Domain Adaptation with Domain Mixup, Dual Mixup Regularized Learning for Adversarial Domain Adaptation. Besides the difference in application, how is XmixUp different from these cross-domain mixup strategies? Please explain on this.

Authors compare few existing method in experiments. However, there are few very related recent methods that should be compared in the experiments to verify the effectiveness of the proposed approach. E.g., Why comparison with Delta: Deep learning transfer using feature map with attention for convolutional networks, ICLR 2019 is missing in the paper? Although DELTA is based on regularization, it is one of the recent transfer learning approach that has much better results compared to L2-SP. Comparison with recent baselines are essential to show the advantage of the proposed approach in transfer learning.

How is the proposed method different from Learning to Transfer Learn: Reinforcement Learning-Based Selection for Adaptive Transfer Learning, ECCV 2020? L2TL reweighs source samples in a multi-task learning set up which is very close to the current approach. Authors should compare and discuss L2TL in the paper.

How is the proposed method comparable to Pay Attention to Features, Transfer Learn Faster CNNs, ICLR 2020? AFDS is much better than L2SP and should be compared in the experiments to verify the effectiveness of the proposed transfer learning approach.

I would like the authors to do more works related to experiments to improve the quality of the paper. I think the paper needs significant changes including new experiments and discussions before being accepted to any major conference.

---

> ### Author Response · Authors · 2020-11-25
> **Thank you for your comments**
>
> Thank you for your review and constructive comments.  We believe all your comments could greatly help us improve the manuscript.  Here we would like to clarify some major concerns in your comments.
>
> 1. The motivation and technical novelty of the method.
>
> This work is motivated by inventing a practical transfer learning strategy, aiming at simultaneously being effective and efficient. Main stream ideas include two kinds, which are (1) regularized learning methods: incorporating regularizers to preserve the knowledge learned by the source model, and (2) multitask learning methods: borrowing relevant examples from the source task and learning together (or sequentially) with the target task. Intuitively, multitask learning methods should have higher potentials as the auxiliary dataset contains the original and complete knowledge about the source task. While current transfer learning algorithms are far from being satisfied especially for their inefficiency. Our work intends to tackle this problem by incorporating Mixup. Serving as a simple and effective training technique, Mixup has been successfully applied to many related machine learning tasks such as domain adaptation and semi-supervised learning. From the perspective of introducing a successful approach of transfer learning, incorporating fine-tuning with subset selection and Mixup makes sense and needs validation. While from the perspective of algorithm invention, these techniques are actually not novel just as review#1 criticizes.
>
> 2. How is XMixup different with Mixup used in domain adaptation.
>
> According to the taxonomy of transfer learning, fine-tuning (inductive transfer learning) and domain adaptation belong to different categories for their different problem settings. In fine-tuning, the auxiliary examples to be mixed are selected from the source task, which have a different label space with the target task. But in domain adaptation, the source and target domain have the same label space. Adversarial training is widely used in domain adaptation to generate domain invariant representations (a discriminator is learned to distinguish between the two domains). In inductive transfer learning tasks, two domains can be concurrently learned as they both are fully labeled.
>
> 3. Comparison with more transfer learning methods.
>
> We do not compare many transfer learning methods as our paper intend to introduce an efficient method to utilize auxiliary examples (from the source dataset). Therefore, we mainly compare with multitask learning methods which adopt other manners to incorporate the auxiliary dataset, e.g. co-learning in a multi-task manner and sequential learning. The ECCV’2002 paper focuses on an advanced strategy to select auxiliary samples, in which we do not deeply investigated. The ICLR’2019 and ICLR’2020 papers propose novel regularizers based on similar ideas of L2-SP that they try to accurately preserve the knowledge of the pre-trained model. For regularization methods, we only compare with L2-SP and BSS because we think it’s enough to show state-of-the-art multitask learning based fine-tuning algorithms are comparable with regularization methods. In other words, the experiments on transfer learning regularizers are conducted to verify that multitask learning based methods are among advanced techniques and thus are useful. We argue that a useful algorithm is expected to have its particular value, but don’t agree that a new algorithm must outperform all previous ones which are designed from various perspectives. While we admit that our algorithm should still have the potential to improve. It will be more exciting if the mechanisms underlying the effectiveness are clearly exploited.

---

### Decision · Program_Chairs · 2021-01-07
**Final Decision**

**Decision:**

Reject

**Comment:**

All reviewers recommend rejection due to limited novelty and insufficient experimental analysis. The author’s response has addressed several other questions raised by the reviewers, but it was not sufficient to eliminate the main concerns about novelty (as the method is a combination of existing techniques) and missing comparisons to justify the effectiveness of the proposed approach.